# Genetic Diversity and Population Structure in Türkiye Bread Wheat Genotypes Revealed by Simple Sequence Repeats (SSR) Markers

**DOI:** 10.3390/genes14061182

**Published:** 2023-05-29

**Authors:** Aras Türkoğlu, Kamil Haliloğlu, Seyyed Abolgahasem Mohammadi, Ali Öztürk, Parisa Bolouri, Güller Özkan, Jan Bocianowski, Alireza Pour-Aboughadareh, Bita Jamshidi

**Affiliations:** 1Department of Field Crops, Faculty of Agriculture, Necmettin Erbakan University, 42310 Konya, Turkey; aras.turkoglu@erbakan.edu.tr; 2Department of Field Crops, Faculty of Agriculture, Ataturk University, 25240 Erzurum, Turkey; aozturk@atauni.edu.tr (A.Ö.);; 3Department of Plant Breeding and Biotechnology, Faculty of Agriculture, University of Tabriz, Tabriz 5166616471, Iran; mohammadi@tabrizu.ac.ir; 4Department of Biology, Faculty of Science, Ankara University, 06100 Ankara, Turkey; gullerozzkan@gmail.com; 5Department of Mathematical and Statistical Methods, Poznań University of Life Sciences, Wojska Polskiego 28, 60-637 Poznań, Poland; 6Seed and Plant Improvement Institute, Agricultural Research, Education and Extension Organization (AREEO), Karaj 31585-854, Iran; a.poraboghadareh@edu.ikiu.ac.ir; 7Department of Food Security and Public Health, Khabat Technical Institute, Erbil Polytechnic University, Erbil 44001, Iraq; bitaalimer@epu.edu.iq

**Keywords:** bread wheat, gene diversity, SSR, population structure

## Abstract

Wheat genotypes should be improved through available germplasm genetic diversity to ensure food security. This study investigated the molecular diversity and population structure of a set of Türkiye bread wheat genotypes using 120 microsatellite markers. Based on the results, 651 polymorphic alleles were evaluated to determine genetic diversity and population structure. The number of alleles ranged from 2 to 19, with an average of 5.44 alleles per locus. Polymorphic information content (PIC) ranged from 0.031 to 0.915 with a mean of 0.43. In addition, the gene diversity index ranged from 0.03 to 0.92 with an average of 0.46. The expected heterozygosity ranged from 0.00 to 0.359 with a mean of 0.124. The unbiased expected heterozygosity ranged from 0.00 to 0.319 with an average of 0.112. The mean values of the number of effective alleles (Ne), genetic diversity of Nei (H) and Shannon’s information index (I) were estimated at 1.190, 1.049 and 0.168, respectively. The highest genetic diversity (GD) was estimated between genotypes G1 and G27. In the UPGMA dendrogram, the 63 genotypes were grouped into three clusters. The three main coordinates were able to explain 12.64, 6.38 and 4.90% of genetic diversity, respectively. AMOVA revealed diversity within populations at 78% and between populations at 22%. The current populations were found to be highly structured. Model-based cluster analyses classified the 63 genotypes studied into three subpopulations. The values of *F*-statistic (*Fst*) for the identified subpopulations were 0.253, 0.330 and 0.244, respectively. In addition, the expected values of heterozygosity (He) for these sub-populations were recorded as 0.45, 0.46 and 0.44, respectively. Therefore, SSR markers can be useful not only in genetic diversity and association analysis of wheat but also in its germplasm for various agronomic traits or mechanisms of tolerance to environmental stresses.

## 1. Introduction

Bread wheat (*Triticum aestivum* L.) is one of the most important species belonging to the genus *Triticum* in the Poaceae family [1]. The genomic structure (2n = 6x = 42, AABBDD) of this cereal consisted of three diploid genomes AA, BB and DD, which are inherited from three ancestral species—*Triticum urartu* Thuman ex Gandil (A genome), *Aegilops speltoides* Tausch (B genome) and *Aegilops tauschii Coss* (DD genome) [2]. The rather large genome size (17,000 Mb) and the high rate of repetitive sequences (80%) are important issues to overcome in bread wheat research [3]. Therefore, efficient and sufficient tools should be used in bread wheat genome research.

The green revolution has resulted in increased yield and quality in wheat production and emergence of high-yield varieties. As the world’s population continues to grow, climate change and the resulting global warming are having a serious impact on food supplies. World wheat production is expected to increase by about 50% by 2050 to meet the food needs of a growing population [4,5]. However, in recent decades, wheat yields have not been able to increase sufficiently in the world [6], as well as in Türkiye [7]. As a result, wheat production is unable to meet demand. Given the negative effects of climate change and the growing world population, which is expected to exceed 9 billion by 2050, the need to increase wheat production to ensure global food security is a high priority [8]. In this case, the biggest challenge for wheat farmers is to improve grain yields and crop tolerance to various environmental stressors to meet growing demands [9]. Wheat has accumulated quite a large amount of genetic variability during its evolution. Today, such a large amount of genetic diversity has generally decreased due to repeated cultivation, adaptation, development and use of local varieties for desirable traits [10]. However, the increased homogeneity of the genetic background has become a major challenge for the future genetic development of wheat.

Plant breeding programs mainly focus on genetic diversity, inheritance, conservation and evolution [11]. Homogeneity in a population would mean that all members of that population behave similarly in the face of a stressor and could not withstand an epidemic [12]. Potential new alleles can be used to overcome such adverse conditions [13]. Genetic diversity is a key topic for the adaptation and survival of wheat species to biotic and abiotic stressors, as such stressors are expected to be major constraints to food security [14]. On the other hand, domestication and selection pressures, as well as the use of modern breeding techniques, have already narrowed the wheat gene pool [15]. National and regional strategies should be developed to characterize and preserve the genetic diversity of wheat species. The decline in the level of genetic diversity has led to the use of such genetic resources in breeding programs. Morphological and molecular markers are commonly used to characterize wheat species and assess genetic diversity. Such tools allow breeders to select genotypes that are well adapted to specific conditions and resistant to various biotic and abiotic stresses. Agromorphological markers, special quantitative traits, are often influenced by environmental factors. To address this problem, several molecular markers have emerged as biotechnological tools for studying genetic diversity and population structure [16]. With the development of biological aspects, a number of molecular marker techniques have emerged, such as random amplified polymorphic DNA (RAPD) [17], amplified fragment length polymorphisms (AFLP) [18], inter-simple sequence repeats (ISSR) [19], start codon targeted markers (SCoT) [20], Inter-primer binding site (iPBS)-retrotransposons [21], expressed sequence tag (EST) [22], single nucleotide polymorphism (SNP) [23], next-generation sequencing (NGS) [24], divergence array technology (DArT) [25] and simple sequence repeats (SSR) [26] have been developed. Of these, SSR markers served as effective molecular markers for studying genetic diversity in hexaploid Türkiye wheat embryos [7,27,28]. The number of genotypes and markers used in these studies seemed insufficient for genome-wide association mapping. SSR markers play a key role in marker-assisted selection (MAS) in wheat breeding programs [29]. Currently, SSR databases are available for various crops [30]. To date, many studies have identified SSR markers as effective tools for use in breeding programs [31]. It has been reported that SSR markers offer a more efficient choice than SNPs, due to their faster mutation rates and higher levels of polymorphism that can be found with several highly polymorphic markers [32]. Therefore, SSR markers are largely used to analyze genetic diversity and population structure, as well as to elucidate phylogenetic relationships among plant genetic resources, as such relationships play a key role in developing appropriate breeding programs [30]. SSR markers can originate from coding or non-coding regions of genomes [33]. It was previously reported that these markers located in promoter regions can affect gene expression levels, while those located in coding sequences can affect protein structure and function [34]. SSR markers have many advantages, such as co-dominance, high levels of polymorphism, chromosome specificity and high reproducibility; they are also excellent for identifying and monitoring target traits within varieties [35]. SSR markers are very efficient in wheat research due to their co-dominant structure and wide coverage across the genome [29].

Türkiye encompasses a high level of bread wheat genetic diversity as it is a major center of wheat domestication and diversity. However, there is little information on the population structure and germplasm diversity of wheat. Therefore, the main objective of this study was to investigate genetic diversity and population structure in a set of Türkiye wheat genotypes using SSR markers.

## 2. Materials and Methods

### 2.1. Genetic Materials

In this study, 63 genotypes of bread wheat (*Triticum aestivum* L.) were used as plant material. Variety names and locations are given in Table 1. Bread wheats were collected from eight different regions of Türkiye. All samples were obtained from the Türkiye National Gene Bank [36].

### 2.2. Extraction of Genomic DNA

Genomic DNA extractions were performed according to the CTAB protocol [37]. The quality of extracted DNA was assessed by agarose gel electrophoresis (0.8%).

### 2.3. PCR Amplification

For SSR analysis, a total of 425 SSR primers were tested on five randomly selected wheat genotypes. Of the primers tested, 120 polymorphic primers were selected for PCR amplification in all 63 wheat genotypes [38]. Subsequently, 120 out of 425 SSR markers were selected for genotyping all 63 sets of bread wheat. Details of the primers used in this study are given in ST1. 

A thermocycler (SensoQuest Labcycler, Göttingen, Germany) was used for PCR amplification. PCR reactions were carried out in a volume of 10 µL and consisted of 25 ng template DNA, 0.5 U Taq polymerase, 0.25 mM dNTP, 1 µM (20 pmol) primer, 10× buffer and 2 mM MgCl_2_. PCR procedure included: 3 min initial denaturation at 95 °C, 38 cycles at 95 °C for 60 s, 50–60 °C (annealing temperature depending on primers for details, see Appendix A) for 60 s, 120 s at 72 °C and final elongation at 72 °C for 10 min [39]. PCR products were stained with 1 µg/mL ethidium bromide and separated by polyacrylamide Mega-Gel dual vertical electrophoresis (Model C-DASG-400-50). The resulting banding pattern was visualized under UV light using a digital camera (Model Nikon Coolpix500, Nikon, Japan) [40].

### 2.4. Statistical Data Analysis

TotalLab TL120 software (TotalLab Ltd., Gosforth, Newcastle upon Tyne, UK) was used to generate matrices [41]. Several informative parameters such as major allele frequency (MAF), gene diversity (GD) and polymorphic information content (PIC) were estimated using Power Marker version 3.25. POPGEN1.32 software was used to determine unbiased expected heterozygosity (uHe), expected heterozygosity (Exp-Het), effective number of alleles (ne), expected heterozygosity Nei (h) and Shannon’s information index (I) values [42]. The Dice similarity index [43] was used to calculate the genetic similarity between each pair of genotypes. NTSYS-pc V2.1 was used to construct a dendrogram using the unweighted double group method with arithmetic mean (UPGMA) and SAHN clustering [25]. Principal coordinate analysis (PCoA) and molecular analysis of variance (AMOVA) were calculated using GenAlExV6.5 [44]. A clustering algorithm on the Bayesian model STRUCTURE 2.2 was used to obtain an explicit picture of genetic composition [45]. For this analysis, input values and parameters were selected as described by Evanno et al. [45]. Finally, the number of actual sub-populations was determined using the Structure Harvester website [46]. MCMC chains were run with a firing period of 100,000 iteration, followed by 100,000 iterations using a model that allowed for admixture and correlated allele frequencies.

## 3. Results

### 3.1. Marker Polymorphism, Genetic Diversity and Principal Coordinate Analysis (PCoA)

Information on the descriptive parameters of SSR markers is shown in Table 2. Of the 425 markers, 120 showed polymorphisms. Genetic variation in SSR loci of bread wheat genotypes was calculated based on Na, MAF, Exp-Het, uHe, GD, H, NE, I and PIC values (Table 2). It was confirmed that 120 SSR loci had a total of 651 alleles in 63 wheat genotypes. The number of alleles per polymorphic locus varied between 2. 00 (BARC 37, BARC 64, BARC 80, BARC 88, BARC 89, BARC 94, BARC 152, BARC 175, BARC 240, CFA, 2, CFA 2187, CFA 152, CFA 2070, CFA 2099, CFD 18, CFD 190, GWM 160, GWM 340, GWM 391, GWM 443, WMC 42, WMC 261, WMC 320, WMC 333, WMC 336, WMC 420, WMC 524, WMC 580, WMC 765, WMC 805, WMC 807, WMC 173 and WMS 72) and 19. 0 (WMC 500) with an average value of 5.442. MAF values ranged from 0.143 (WMC 500) to 0.984 (BARC 37, BARC 80, BARC 88, BARC 89, BARC 94, BARC 152, CFA 2187, CFD 18, GWM 340, WMC 261, WMC 320, WMC 333, WMC 524 and WMC 807) with an average value of 0.631 (Table 2).

Exp-He ranged from 0.00 (BARC 37, BARC 94, CFA 2070 and GWM 340) to 0.359 (GWM 350) with a mean of 0.124. uHe values ranged from 0.00 (BARC 37, BARC 94, CFA 2070 and GWM 340) to 0.319 (GWM 350) with a mean of 0.112 (Table 2). GD values ranged from 0.031 (BARC 37) to 0.920 (WMS 46) with a mean value of 0.460. The highest Ne, H and I values were 1.578 (GWM 350), 1.667 (WMC 687) and 0.459 (GWM 350), respectively, while the lowest Ne, H and I values were 1. 00 (BARC 37, BARC 94, BARC 206, CFA 2070, GWM 340), 0.556 (BARC 206) and 0.00 (BARC 37, BARC 94, BARC 206, CFA 2187 and WMC 173) with mean values of 0.110, 1.187 and 0.165, respectively. PIC values ranged from 0.031 (BARC 37, BARC 80, BARC 88, BARC 89, BARC 94, BARC 152, CFA 2187, CFD 18, GWM 340, WMC 261, WMC 320, WMC 333, WMC 524, WMC 807) to 0.915 (WMS 462) with a mean value of 0.430 (Table 2).

Principal coordinate analysis (PCoA) was conducted using Nei’s neutral genetic distance. The three principal coordinates explained 12.64, 6.38 and 4.90% of genetic diversity, respectively (23.92% diversity in total). The presence of genetic diversity was confirmed by the distribution of genotypes in the diagram (Figure 1). The results of the AMOVA showed that the fraction of genetic diversity within populations was greater than between them (78% vs. 22%) (Table 3).

### 3.2. Genetic Distance and Cluster Analysis for SSR Markers 

Phylogenetic relationships were investigated for 63 bread genotypes using 120 SSR markers. Dice similarity coefficients were calculated for the 120 SSR markers, and a UPGMA tree was generated (Figure 2). Genetic diversity (GD) values ranged from 0.184 to 0.420 with a mean of 0.279. The highest GD was observed between genotypes G1 and G27, while the lowest was found between four wheat samples (G10 and G11; G41 and G; G43 and G; G62 and G63). The wheat genotypes were grouped into three main clusters. The first cluster (Cluster I) was divided into two sub-clusters. The first sub-cluster (G I-1) included 29 genotypes (G63, G62, G60, G59, G58, G57, G56, G61, G55, G54, G53, G52, G50, G49, G51, G48, G46, G45, G44, G47, G42, G43, G41, G40, G39, G38, G37, G36 and G35). The second sub-cluster (G II-2) included 27 genotypes (G27, G26, G29, G34, G33, G32, G31, G30, G28, G25, G24, G23, G22, G21, G20, G19, G18, G17, G16, G15, G14, G10, G13, G9, G12, G11 and G6). The second main cluster (Cluster II) had two sub-clusters, the first sub-cluster (G II-1) including four genotypes (G2, G4, G5 and G3) and the second sub-cluster (G II-2) including two genotypes (G7 and G8). The third main cluster (Cluster III) had only one genotype (G1). It was observed that most genotypes were collected within Cluster I (Figure 2).

### 3.3. Population Genetic Structure Analysis for SSR Markers

The results of STRUCTURE analysis classified all tested genotypes into three sub-populations (sub-population A—red, sub-population B—green and sub-population C—blue) with a membership probability as <0.8 (Figure 3). Accordingly, sub-population A included 18 wheat genotypes (28.57%, G20, G21, G30, G23, G26, G24, G22, G13, G28, G25, G15, G16, G18, G14, G29, G31, G19 and G27). Sub-population B included 28 wheat genotypes (44.44%, G53, G57, G58, G56, G40, G52, G54, G60, G43, G45, G47, G49, G55, G50, G51, G48, G61, G46, G42, G59, G41, G44, G63, G39, G37, G62, G38 and G36). Sub-population C included 11 wheat genotypes (17.46%, G4, G2, G5, G3, G6, G1, G9, G7, G11, G10 and G8). In addition, six wheat genotypes (9.52%) including G33, G32, G17, G34, G35 and G12, were placed in mixed groups. The values of the *F*-statistic (*Fst*) for the first, second and third sub-populations were estimated to be 0.253, 0.330 and 0.244, respectively. The expected heterozygosity (He) was determined as 0.452 for the first, 0.463 for the second and 0.444 for the third sub-population (Table 4 and Table 5). Accordingly, sub-populations A and C were identified as the most diverse populations (Table 6).

## 4. Discussion

Detection of genetic variation using molecular markers is highly dependent on the mode of reproduction, domestication history and size of the samples analyzed. Collection, conservation and management of genetic resources are key issues in sustainable agriculture development [47]. Assessing levels and patterns of genetic diversity allows accurate classification of species and identification of individuals with desirable traits [48]. Existing genetic resources, their geographic location and relationships are commonly used to determine population diversity [49]. Comprehensive knowledge of bread wheat genetic diversity will have a significant impact on germplasm conservation and utilization. Such knowledge also facilitates breeding programs. Breeders have made significant progress in detecting various morphological traits and variation of molecular traits at the DNA level [50]. Molecular markers offer efficient tools for improving traditional breeding programs because they are not affected by environmental and developmental factors [51]. SSR markers are commonly used to analyze the genetic diversity of wheat genotypes [30]. In this study, 120 SSR markers were used to determine molecular variation and population structure in core-collection of Türkiye bread wheat genotypes.

### 4.1. Monitoring of Genetic Diversity

Using 120 SSR markers, 651 alleles were identified in 63 wheat genotypes. The number of polymorphic alleles ranged from 2.00 to 19.0 with an average of 5.442. Polymorphism can result from SSR expansion, contraction or interruption [52]. The current mean of polymorphic alleles was higher than the 458 [53], 49 [54] and 38 [31] values, the lower than the 1620 [48] and 939 [27] values represented in previous studies. Teshome et al. [35] reported that the number of alleles (Na) per locus ranged from 2 to 6. The current average number of alleles was lower than the values of 5.7 [55], 10 [56], 10.06 [30], 7.97 [48], 7.2 [57], 6.8 [58], 5.9 [7] and 5.89 [59] and higher than the values of 3.3 [60] and 5.05 [61] reported in previous studies. The differences in the results of these studies are mainly attributed to differences in genotypes and number of markers. The number of alleles per marker largely depends on the relative distance of the locus from the centromere, the allele frequency motif and the number of repeats [16]. Allelic diversity is also influenced by genetic composition, designating the number of alleles per locus [30].

Exp-He values ranged from 0.00 to 0.359 with an average of 0.124. The differences in Obs-He values can be attributed to several factors, including the molecular markers used, the number of selections and the geographic location of the wild–type origin and location of the samples. Our result is higher than that of Teshome et al. [62] with 0–0.05 and lower than that of Ateş Sönmezoğlu [7] with an average value of 0.75 and Tsonev et al. [63] with an average of 0.185.

Genetic diversity (GD) values ranged from 0.031 to 0.920, with an average value of 0.460. Arystanbekkyzy et al. [64] indicated that genetically distinct genotypes can facilitate breeding programs for desired traits. Henkrar et al. [65], Ateş Sönmezoğlu and Terzi [7] and Belete et al. [30] observed greater gene diversity for primers producing a higher number of alleles. Our result was lower than Tsonev et al. [63] with an average of 0.658 and Mohi-Ud-Din [53] with an average of 0.936.

In this study, the highest value of h, ne and I with 1.667, 1.578 and 0.459, respectively, were observed, while the lowest values of h, ne and I were 0.556, 1.00 and 0.00, respectively. A higher number of effective alleles indicates greater genetic diversity and is therefore generally desirable in breeding programs. The Shannon information index is also an indicator of genetic variation in a population. Teshome et al. [66] reported I values with 0.53. These values were greater than the current results. Mohi-Ud-Din et al. [53] reported the average number of effective alleles per locus as 18.32, indicating considerable diversity in the genotypes studied. The lower values of diversity indices in the present study were attributed to differences in germplasm.

PIC and MAF values indicate significant genetic variability among all wheat species used. They are also reliable indicators of genetic diversity in the plant. The current MAF values ranged from 0.143 to 0.984 with an average of 0.631. Our result was higher than Mohi-Ud-Din et al. [53] with an average of 0.296. The polymorphism information content (PIC) is used as an indicator of the diversity of a gene or DNA segment of a population. It also indicates evolutionary pressure on alleles and mutations. Current PIC values ranged from 0.031 to 0.915 with an average of 0.430. In this study, 25 markers had a PIC value of ≥0.5, indicating their potential use in wheat germplasm genetic diversity studies. Locus has high diversity when the PIC value is ≥0.5 and low diversity when the PIC value is ≤0.25 [67]. In similar studies conducted on wheat genotypes with SSR markers, the average PIC values were lower than the value of 0.62 [63], 0.65 [68], 0.65, [48], 0.52 [28], 0.50 [7], 0.57 [69], 0.83 [53] and higher than Erayman et al. [27] with an average of 0.205, Demirel [70] with an average of 0.19 and Pour-Aboughadareh et al. [54] with an average of 0.32, and Kumar et al. [51] reported an average of 0.33 PIC values. Of the 25 SSR markers, 21 had a PIC value greater than 0.800, indicating that these markers were highly informative and effective.

Principal coordinate analysis (PCoA) is commonly used to spatially represent relative genetic distances between populations [57]. It is also a multidimensional dataset that provides key patterns across multiple loci and samples. The two-dimensional diagram reflects the actual distances between genotypes [56]. In this study, the three main coordinates were able to explain 12.64, 6.38 and 4.90% (23.92% in total) of the total variation. Data were considered reliable when the explained portion of variation was ≥25% [71]. SSR-based clustering offers reliable differentiation of wheat genotypes based on their origin. In this study, significant correlations were found between PCoA clustering and cluster analysis. Mohi-Ud-Din et al. [53] indicated that PCoA was unable to group 56 genotypes based on their population. However, Pour-Aboughadareh et al. [54] found that PCoA confirmed the clustering pattern. Based on AMOVA results, there was more variability within populations (78%) than between populations (22%). Consistent with our results, Mohi-Ud-Din et al. [53] found that differences between populations accounted for 7% of total genetic diversity, with the rest (93%) attributed to differences within populations.

### 4.2. Genetic Identity, Genetic Distance and Clustering Anlaysis 

Genetic differences between populations play a huge role in the conservation of genetic resources [72]. Our results showed that the highest genetic distance (GD) occurred between G1 and G27 and the lowest between G10 and G11; G41 and G; G43 and G; G62 and G63. Kumar et al. [12] found dissimilarity indices ranging from 0.62 to 0.85. Erayman et al. [27] reported similarity indices between 0.52 and 0.97 for all species and between 0.69 and 0.97 for wheat cultivars.

The current SSR markers were able to group all genotypes well based on phylogenetic relationships. The UPGMA method divided the present genotypes into three main clusters. Cluster I included 56 (88.88%); cluster II included 6 (9.52%); and cluster III included 1 (1.58%) genotype. UPGMA analysis showed a mix of frequencies as submissions from different geographic regions were grouped into the same subgroups. The current results showed that the clustering models were not able to clearly distinguish between wheat genotypes based on geographic origin. Clustering of genotypes showed no significant relationship between geographic origin and genetic similarity. Such a case indicated gene flow between genotypes. Differences between genotypes were attributed to the greater genetic distance between them. Grouping based on geographic origin was not clear. Such findings were also supported by analysis of population structure. The present results are consistent with those of Mohammadi et al. [71] and Pour-Aboughadareh et al. [54]. Tsonev et al. [63] divided 117 varieties into 2 major clusters, consistent with the 2 major subpopulations of the K = 2 genetic structure analysis. Mohi-Ud-Din et al. [53] used UPGMA analysis to assess the genetic diversity of wheat genotypes using SSR markers and grouped wheat genotypes into five major clusters. Pour-Aboughadareh et al. [54] found that for phylogenetic relationships, SSR markers gave better performance than gene-based techniques.

### 4.3. Population Diversity, Gene Differentiation and Gene Flow of Populations

Natural diversity is used to analyze population structure to detect genes/qTLs of agronomic traits [73]. Such analysis reveals similarities between genotypes and sub-populations. It has been proven to be more reliable and provide more information than other clustering algorithms [30]. The population structure facilitates the selection of different parents and the mapping of marker–trait relationships for use in breeding programs. In this study, analysis of the population structure showed that all varieties came from three subpopulations. The genetic composition of a population is largely determined by various factors, including recombination, genetic drift and natural selection. Subpopulation A contained 18 wheat genotypes (28.57%); subpopulation B contained the highest number of genotypes (28–44.44%); and subpopulation C contained 11 wheat genotypes (17.46%). In addition, 6 wheat genotypes (9.52%) were in mixed groups.

The smallest number of genotypes included in the mixed groups indicated that the genotypes present had a wide range of genetic pools. This study analyzed the population structure of wheat genotypes representing the diversity of Türkiye wheat genotypes. The Bayesian model yielded similar clustering results to UPGMA and PCoA. STRUCTURE analysis revealed three groups (A, B and C) at K = 3. Group B contained the highest number of genotypes. The present results on population structure are consistent with the findings of Mohi-Ud-Din et al. [53], dividing 56 wheat genotypes into three sub-populations, as well as Tascioglu et al. [48], dividing wheat genotypes into three sub-groups based on Bayesian model and PCA. On the other hand, the present results on population structure are not consistent with those of Le Couviour et al. [74] and Tsonev et al. [63], mainly due to the different genetic materials used in these studies. *F*-statistic (*Fst*) value was determined to be 0.253 for the first, 0.330 for the second and 0.244 for the third sub-population. Th expected value of heterozygosity (He) was determined as 0.452 for the first, 0.463 for the second and 0.444 for the third sub-population. Mohi-Ud-Din et al. [53] reported two significant differences (*p* < 0.01) in paired population *Fst* values.

## 5. Conclusions

To facilitate the conservation, classification and maintenance, as well as the use of these valuable genes available in genetic resources, genetic diversity analysis is needed. In Türkiye, many efforts have been made to identify the best wheat genotypes in terms of yield and agromorphological traits. Although wheat genotypes collected from some regions have been previously characterized using other marker systems, here the SSR marker set was used to assess genetic diversity and population structure in set of bread wheat genotypes. Our results showed acceptable values for average allele number, PIC, GD, Ex-He, u-He parameters. In addition, the mean values of Ne, H and I for all genotypes tested were estimated at 1.190, 1.049 and 0.168, respectively. AMOVA showed that variability within populations was higher than between them (78% vs. 22%). In addition, the *Fst* values for the assumed sub-populations were 0.253, 0.330 and 0.244, respectively. In conclusion, our findings again showed that there is a high level of genetic diversity among Türkiye bread wheat genotypes, which in turn can be taken into account in future wheat breeding programs.

## Figures and Tables

**Figure 1 genes-14-01182-f001:**
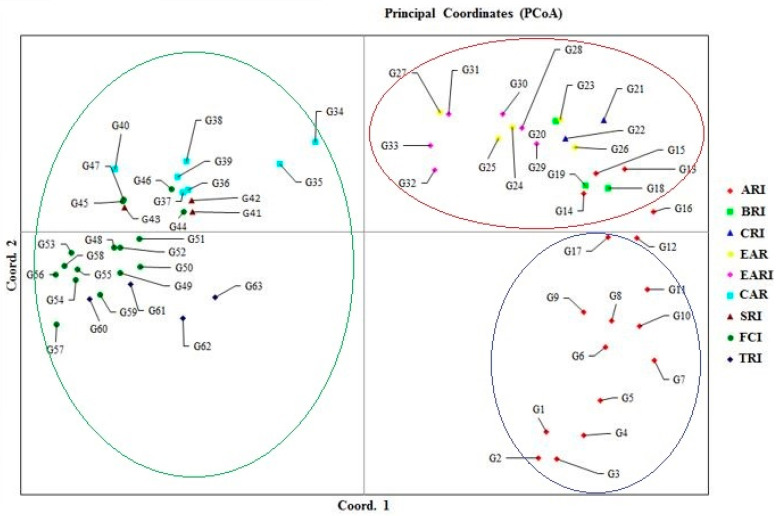
Biplot rendered using PCoA analysis for 63 Türkiye bread wheat genotypes based on SSR marker data. Ari: Anatolia Agricultural Research Institute, Bri: Bahri Dagdas International Agricultural Research Institute, Cri: Çukurova Agricultural Research Institute, Ear: Eastern Anatolia Region, Eari: Eastern Anatolia Agricultural Research Institute, Car: Central Anatolia Region, Sri: Sakarya Agricultural Research Institute, Fci: Field Crops Central Research Institute, Tri: Trakya Agricultural Research Institute indicate, respectively.

**Figure 2 genes-14-01182-f002:**
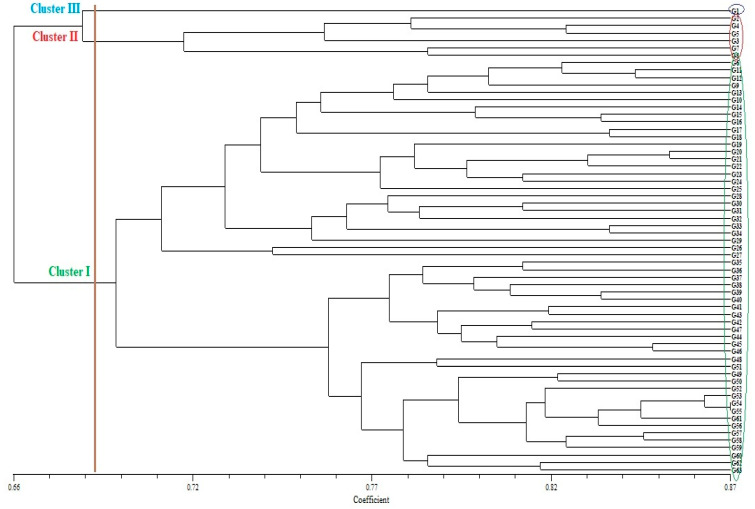
The UPGMA dendrogram shows the grouping of 63 Türkiye bread wheat genotypes based on 120 SSR markers data.

**Figure 3 genes-14-01182-f003:**
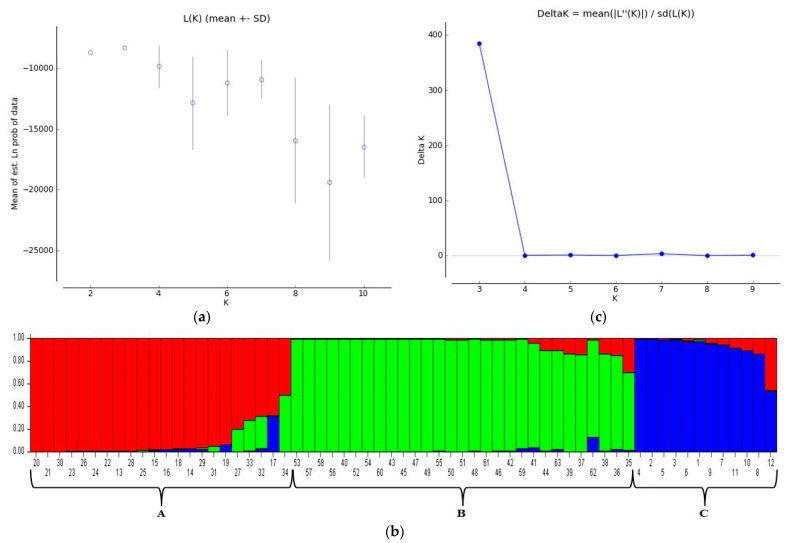
Estimation of number of groups. (**a**) Plot of ΔK over K (range 2–10), (**b**) Bar plot grouping of 63 genotypes, (**c**) Line graphs from the mixture model of Ln P (D) and ∆K.

**Table 1 genes-14-01182-t001:** The passport of the investigated Türkiye bread wheat genotypes.

Code	Genotype/Variety Name	Variety Owner Organization/Origin	Season of Sowing
G1	Aksel 2000	Field Crops Central Research Institute	Facultative
G2	Alparslan	Eastern Anatolia Agricultural Research Institute	Winter
G3	Altay 2000	Anatolia Agricultural Research Institute	Winter
G4	Atlı 2002	Field Crops Central Research Institute	Facultative
G5	Aytın 98	Anatolia Agricultural Research Institute	Winter
G6	Bağcı 2002	Bahri Dağdaş International Agricultural Research Institute	Facultative
G7	Bayraktar 2000	Field Crops Central Research Institute	Facultative
G8	Bolal 2973	Anatolia Agricultural Research Institute	Facultative
G9	Çetinel 2000	Anatolia Agricultural Research Institute	Winter
G10	Dağdaş 94	Bahri Dağdaş International Agricultural Research Institute	Facultative
G11	Demir 2000	Field Crops Central Research Institute	Facultative
G12	Doğankent 1	Çukurova Agricultural Research Institute	Spring
G13	Doğu 88	Eastern Anatolia Agricultural Research Institute	Winter
G14	Gerek 79	Anatolia Agricultural Research Institute	Winter
G15	Gün 91	Field Crops Central Research Institute	Winter
G16	Harmankaya 99	Anatolia Agricultural Research Institute	Winter
G17	İkizce 96	Field Crops Central Research Institute	Facultative
G18	İzgi 2001	Anatolia Agricultural Research Institute	Winter
G19	Karahan 99	Bahri Dağdaş International Agricultural Research Institute	Winter
G20	Kate A-1	Trakya Agricultural Research Institute	Winter
G21	Kıraç 66	Anatolia Agricultural Research Institute	Winter
G22	Kırgız 95	Anatolia Agricultural Research Institute	Winter
G23	Kırkpınar 79	Trakya Agricultural Research Institute	Facultative
G24	Kutluk 94	Anatolia Agricultural Research Institute	Winter
G25	Lancer	Eastern Anatolia Agricultural Research Institute	Winter
G26	Mızrak	Field Crops Central Research Institute	Facultative
G27	Müfitbey	Anatolia Agricultural Research Institute	Winter
G28	Nenehatun	Eastern Anatolia Agricultural Research Institute	Winter
G29	Palandöken 97	Eastern Anatolia Agricultural Research Institute	Winter
G30	Pamukova 97	Sakarya Agricultural Research Institute	Spring
G31	Pehlivan	Trakya Agricultural Research Institute	Winter
G32	Prostor	Trakya Agricultural Research Institute	Winter
G33	Seri 82	Çukurova Agricultural Research Institute	Winter
G34	Soyer02	Anatolia Agricultural Research Institute	Winter
G35	Sönmez 2001	Anatolia Agricultural Research Institute	Winter
G36	Sultan 95	Anatolia Agricultural Research Institute	Winter
G37	Süzen 97	Anatolia Agricultural Research Institute	Winter
G38	Tosunbey	Field Crops Central Research Institute	Winter
G39	Türkmen	Field Crops Central Research Institute	Facultative
G40	Uzunyayla	Field Crops Central Research Institute	Facultative
G41	Yakar 99	Field Crops Central Research Institute	Facultative
G42	Zencirci 2002	Field Crops Central Research Institute	Facultative
G43	Ak-702	Anatolia Agricultural Research Institute	Winter
G44	Ak buğday	Central Anatolia Region	Winter
G45	Ankara 093/44	Field Crops Central Research Institute	Winter
G46	Conkesme	Eastern Anatolia Region	Facultative
G47	Haymana 79	Field Crops Central Research Institute	Winter
G48	Kılçıksız buğday	Central Anatolia Region	Winter
G49	Kırik	Eastern Anatolia Region	Facultative
G50	Kırmızı Kılçık	Eastern Anatolia Region	Facultative
G51	Kırmızı Yerli	Eastern Anatolia Region	Facultative
G52	Koca buğday	Central Anatolia Region	Winter
G53	Köse 220/39	Field Crops Central Research Institute	Facultative
G54	Orso	Sakarya Agricultural Research Institute	Facultative
G55	Özlü buğday	Central Anatolia Region	Winter
G56	Polatlı Kösesi	Central Anatolia Region	Facultative
G57	Sert buğday	Central Anatolia Region	Winter
G58	Sürak 1593/51	Field Crops Central Research Institute	Winter
G59	Tir	Eastern Anatolia Region	Winter
G60	Yayla 305	Anatolia Agricultural Research Institute	Winter
G61	Zerin	Central Anatolia Region	Facultative
G62	Bezostaja 1	Sakarya Agricultural Research Institute	Winter
G63	Karasu 90	Eastern Anatolia Agricultural Research Institute	Winter

**Table 2 genes-14-01182-t002:** List of the used SSR primers along with the results of estimated informativeness parameters for each of them.

Marker	Na	MAF	Ex-He	u-He	GD	Ne	H	I	PIC
BARC 1	7.00	0.302	0.223	0.201	0.773	1.343	1.259	0.301	0.739
BARC 3	3.00	0.841	0.029	0.027	0.272	1.036	0.630	0.047	0.242
BARC 24	3.00	0.968	0.042	0.041	0.062	1.065	1.111	0.062	0.061
BARC 37	2.00	0.984	0.000	0.000	0.031	1.000	1.000	0.000	0.031
BARC 45	3.00	0.968	0.042	0.041	0.062	1.065	1.111	0.062	0.061
BARC 48	3.00	0.778	0.108	0.096	0.358	1.172	0.889	0.138	0.313
BARC 54	11.0	0.317	0.133	0.121	0.794	1.195	0.870	0.187	0.767
BARC 59	8.00	0.317	0.302	0.267	0.797	1.459	1.583	0.398	0.770
BARC 64	2.000	0.968	0.043	0.042	0.061	1.067	1.111	0.062	0.060
BARC 73	4.00	0.413	0.072	0.065	0.696	1.107	0.704	0.098	0.642
BARC 78	13.0	0.238	0.187	0.169	0.860	1.276	1.095	0.260	0.845
BARC 80	2.00	0.984	0.042	0.041	0.031	1.065	1.111	0.062	0.031
BARC 88	2.00	0.984	0.042	0.041	0.031	1.065	1.111	0.062	0.031
BARC 89	2.00	0.984	0.042	0.041	0.031	1.065	1.111	0.062	0.031
BARC 94	2.00	0.984	0.000	0.000	0.031	1.000	1.000	0.000	0.031
BARC 101	7.00	0.317	0.169	0.151	0.786	1.256	0.972	0.225	0.755
BARC 105	7.00	0.333	0.313	0.279	0.781	1.495	1.556	0.412	0.750
BARC 113	8.00	0.333	0.131	0.122	0.749	1.200	1.028	0.189	0.708
BARC 122	7.00	0.254	0.288	0.250	0.795	1.401	1.593	0.382	0.764
BARC 128	6.00	0.349	0.189	0.173	0.731	1.295	1.167	0.258	0.687
BARC 130	4.00	0.762	0.091	0.087	0.391	1.138	0.944	0.135	0.358
BARC 133	3.00	0.905	0.048	0.044	0.176	1.075	0.815	0.065	0.168
BARC 135	7.00	0.317	0.165	0.147	0.761	1.256	1.044	0.220	0.723
BARC 140	10.0	0.349	0.201	0.177	0.802	1.302	1.167	0.267	0.780
BARC 141	6.00	0.825	0.108	0.098	0.309	1.158	0.861	0.154	0.297
BARC 142	6.00	0.317	0.204	0.185	0.761	1.310	1.167	0.278	0.722
BARC 152	2.00	0.984	0.042	0.041	0.031	1.065	1.111	0.062	0.031
BARC 165	6.00	0.349	0.141	0.130	0.742	1.225	0.917	0.193	0.699
BARC 175	2.00	0.937	0.162	0.144	0.119	1.247	1.167	0.217	0.112
BARC 197	3.00	0.857	0.064	0.060	0.249	1.105	0.889	0.090	0.225
BARC 204	3.00	0.952	0.026	0.025	0.092	1.046	1.056	0.036	0.089
BARC 206	3.00	0.921	0.000	0.000	0.148	1.000	0.556	0.000	0.141
BARC 216	10.0	0.381	0.169	0.150	0.771	1.236	1.056	0.236	0.743
BARC 240	2.00	0.968	0.026	0.025	0.061	1.046	1.056	0.036	0.060
CFA 152	6.00	0.714	0.145	0.135	0.469	1.236	1.000	0.203	0.448
CFA 2040	2.00	0.984	0.000	0.000	0.031	1.000	1.000	0.000	0.031
CFA 2049	2.00	0.667	0.172	0.154	0.444	1.276	1.333	0.225	0.346
CFA 2187	4.00	0.905	0.061	0.055	0.177	1.089	0.630	0.085	0.169
CFA 2043	5.00	0.698	0.203	0.186	0.480	1.343	1.222	0.271	0.448
CFA 2070	2.00	0.921	0.000	0.000	0.146	1.000	1.000	0.000	0.135
CFA 2099	2.00	0.968	0.026	0.025	0.061	1.046	1.056	0.036	0.060
CFA 2155	4.00	0.952	0.029	0.028	0.092	1.041	0.778	0.047	0.091
CFA 2163	8.00	0.317	0.261	0.235	0.785	1.411	1.389	0.348	0.753
CFA 2185	8.00	0.381	0.190	0.172	0.775	1.299	1.139	0.257	0.747
CFA 2190	6.00	0.508	0.105	0.093	0.662	1.140	0.889	0.149	0.618
CFA 2256	3.00	0.937	0.017	0.016	0.120	1.025	0.778	0.026	0.116
CFA 2	2.00	0.857	0.053	0.050	0.245	1.086	0.833	0.076	0.215
CFD 18	2.00	0.984	0.042	0.041	0.031	1.065	1.111	0.062	0.031
CFD 49	5.00	0.413	0.324	0.288	0.703	1.499	1.593	0.425	0.652
CFD 190	2.00	0.889	0.028	0.027	0.198	1.054	1.000	0.038	0.178
CFD 287	13.0	0.190	0.190	0.170	0.884	1.281	1.200	0.260	0.873
GWM 160	2.00	0.794	0.167	0.151	0.328	1.281	1.333	0.215	0.274
GWM 299	4.00	0.937	0.023	0.022	0.121	1.031	0.593	0.039	0.119
GWM 314	4.00	0.730	0.219	0.198	0.426	1.369	1.444	0.283	0.383
GWM 319	3.00	0.889	0.056	0.055	0.201	1.107	1.111	0.076	0.186
GWM 337	6.00	0.460	0.107	0.097	0.588	1.158	0.917	0.150	0.501
GWM 340	2.00	0.984	0.000	0.000	0.031	1.000	1.000	0.000	0.031
GWM 350	7.00	0.270	0.359	0.319	0.813	1.578	1.630	0.459	0.788
GWM 368	5.00	0.714	0.209	0.184	0.459	1.319	1.370	0.278	0.427
GWM 382	8.00	0.333	0.225	0.202	0.787	1.336	1.306	0.307	0.758
GWM 391	2.00	0.762	0.097	0.085	0.363	1.140	1.222	0.126	0.297
GWM 413	5.00	0.571	0.153	0.135	0.601	1.242	1.044	0.197	0.552
GWM 443	2.00	0.968	0.089	0.078	0.061	1.135	0.833	0.116	0.060
GWM 493	3.00	0.540	0.155	0.139	0.602	1.239	1.222	0.210	0.534
GWM 497	6.00	0.540	0.217	0.195	0.634	1.331	1.361	0.294	0.589
GWM 501	3.00	0.730	0.174	0.159	0.402	1.280	1.259	0.233	0.333
GWM 533	7.00	0.365	0.180	0.161	0.718	1.274	1.139	0.243	0.671
WMC 42	2.00	0.571	0.061	0.055	0.490	1.098	0.889	0.082	0.370
WMC 99	10.0	0.556	0.156	0.139	0.654	1.236	0.889	0.210	0.630
WMC 114	6.00	0.508	0.113	0.102	0.647	1.172	0.806	0.153	0.595
WMC 166	3.00	0.587	0.035	0.032	0.509	1.057	0.556	0.046	0.408
WMC 210	8.00	0.302	0.188	0.171	0.750	1.282	1.167	0.264	0.707
WMC 261	3.00	0.841	0.119	0.107	0.272	1.205	0.963	0.153	0.242
WMC 317	2.00	0.984	0.021	0.020	0.031	1.032	1.056	0.031	0.031
WMC 320	7.00	0.413	0.094	0.087	0.750	1.137	0.806	0.136	0.719
WMC 329	2.00	0.984	0.021	0.020	0.031	1.032	1.056	0.031	0.031
WMC 333	2.00	0.984	0.021	0.020	0.031	1.032	1.056	0.031	0.031
WMC 336	2.00	0.968	0.061	0.053	0.061	1.102	0.667	0.075	0.060
WMC 356	3.00	0.492	0.144	0.129	0.568	1.222	1.148	0.196	0.474
WMC 361	4.00	0.524	0.141	0.127	0.618	1.227	0.852	0.186	0.554
WMC 406	7.00	0.635	0.160	0.140	0.543	1.236	0.844	0.209	0.500
WMC 413	3.00	0.968	0.049	0.047	0.062	1.082	1.111	0.068	0.061
WMC 420	2.00	0.968	0.061	0.053	0.061	1.102	0.667	0.075	0.060
WMC 435	3.00	0.492	0.058	0.055	0.544	1.095	0.889	0.081	0.439
WMC 468	11.0	0.524	0.157	0.142	0.679	1.237	0.958	0.216	0.653
WMC 500	19.0	0.143	0.201	0.178	0.914	1.276	1.272	0.280	0.908
WMC 524	2.00	0.984	0.021	0.020	0.031	1.032	1.056	0.031	0.031
WMC 532	18.0	0.190	0.157	0.142	0.897	1.229	0.986	0.219	0.889
WMC 553	4.00	0.476	0.182	0.165	0.572	1.287	1.222	0.246	0.480
WMC 580	2.00	0.873	0.055	0.051	0.222	1.089	0.833	0.077	0.197
WMC 617	7.00	0.492	0.192	0.173	0.686	1.300	1.111	0.256	0.648
WMC 687	3.00	0.667	0.332	0.292	0.479	1.506	1.667	0.432	0.410
WMC 765	2.00	0.968	0.113	0.102	0.061	1.191	1.222	0.144	0.060
WMC 805	2.00	0.968	0.089	0.078	0.061	1.135	0.833	0.116	0.060
WMC 807	2.00	0.984	0.036	0.034	0.031	1.058	0.667	0.052	0.031
WMC 810	3.00	0.651	0.160	0.138	0.513	1.217	1.037	0.214	0.458
WMC 173	2.00	0.937	0.000	0.000	0.119	1.000	1.000	0.000	0.112
WMS 5	10.0	0.397	0.156	0.143	0.766	1.235	0.956	0.218	0.739
WMS 10	4.00	0.603	0.140	0.131	0.541	1.215	1.333	0.199	0.470
WMS 24	3.00	0.619	0.185	0.165	0.483	1.305	1.389	0.238	0.381
WMS 44	3.00	0.825	0.105	0.097	0.297	1.173	0.833	0.139	0.268
WMS 46	6.00	0.619	0.159	0.146	0.540	1.258	1.074	0.214	0.481
WMS 52	19.0	0.159	0.191	0.166	0.920	1.257	1.111	0.259	0.915
WMS 55	10.0	0.238	0.274	0.238	0.852	1.407	1.311	0.355	0.836
WMS 58	8.00	0.286	0.150	0.134	0.799	1.208	1.000	0.210	0.772
WMS 63	9.00	0.365	0.107	0.096	0.764	1.167	0.685	0.145	0.731
WMS 67	3.00	0.492	0.108	0.100	0.515	1.167	1.167	0.149	0.398
WMS 72	2.00	0.968	0.113	0.102	0.061	1.191	1.222	0.144	0.060
WMS77	11.0	0.286	0.224	0.202	0.826	1.335	1.267	0.307	0.805
WMS 107	4.00	0.460	0.087	0.081	0.663	1.139	0.815	0.121	0.604
WMS 118	8.00	0.492	0.195	0.176	0.681	1.298	1.139	0.267	0.641
WMS 124	18.0	0.159	0.175	0.152	0.918	1.231	1.136	0.241	0.912
WMS 148	10.0	0.302	0.146	0.134	0.825	1.217	0.911	0.206	0.806
WMS 155	8.00	0.413	0.080	0.074	0.736	1.120	0.593	0.115	0.700
WMS 189	12.0	0.159	0.228	0.204	0.892	1.324	1.352	0.317	0.883
WMS 190	9.00	0.460	0.078	0.073	0.720	1.120	0.611	0.113	0.688
WMS 297	5.00	0.365	0.104	0.097	0.708	1.164	0.778	0.145	0.656
WMS 403	15.00	0.238	0.146	0.129	0.867	1.208	0.903	0.201	0.854
WMS 493	12.00	0.270	0.173	0.154	0.826	1.259	0.978	0.233	0.805
WMS 566	4.00	0.651	0.231	0.208	0.531	1.397	1.444	0.294	0.491
Mean	5.442	0.631	0.124	0.112	0.460	1.190	1.049	0.168	0.430

Na: Observed number of alleles, MAF: Major allele frequency, Exp-He: Expected heterozygosity, uHe: unbiased expected heterozygosity, GD: Gene diversity, Ne: Effective number of alleles, H: Nei’s expected heterozygosity, I: Shannon’s Information index, PIC: Polymorphism information content.

**Table 3 genes-14-01182-t003:** Results of AMOVA analysis for investigated Türkiye bread wheat genotypes.

Source of Variation	df	Sum of Squares (SS)	Mean of Squares (MS)	Variance Component	% of Total Variance (%)
Among Population	8	1054.528	131.816	13.026	22
Within Population	54	2546.107	47.150	47.150	78
Total	62	3600.635		60.176	100

**Table 4 genes-14-01182-t004:** Membership coefficients of investigated Türkiye bread wheat genotypes in each estimated sub-population.

Code	Subpopulation				Subpopulation		
	I	II	III	Code Number	I	II	III
G1	0.011	0.017	0.973	G33	0.720	0.272	0.009
G2	0.002	0.004	0.994	G34	0.500	0.497	0.004
G3	0.005	0.009	0.986	G35	0.294	0.685	0.021
G4	0.003	0.002	0.995	G36	0.146	0.830	0.024
G5	0.010	0.003	0.987	G37	0.137	0.859	0.004
G6	0.019	0.003	0.978	G38	0.135	0.857	0.008
G7	0.051	0.003	0.946	G39	0.130	0.864	0.006
G8	0.129	0.006	0.865	G40	0.004	0.994	0.002
G9	0.037	0.008	0.955	G41	0.036	0.922	0.042
G10	0.102	0.003	0.896	G42	0.009	0.977	0.014
G11	0.080	0.002	0.917	G43	0.006	0.992	0.002
G12	0.456	0.002	0.542	G44	0.103	0.887	0.010
G13	0.988	0.002	0.010	G45	0.005	0.992	0.003
G14	0.967	0.004	0.029	G46	0.010	0.982	0.008
G15	0.978	0.003	0.019	G47	0.004	0.992	0.004
G16	0.975	0.003	0.022	G48	0.006	0.986	0.008
G17	0.676	0.004	0.320	G49	0.004	0.992	0.004
G18	0.968	0.004	0.028	G50	0.008	0.987	0.005
G19	0.930	0.003	0.067	G51	0.008	0.987	0.005
G20	0.995	0.003	0.003	G52	0.004	0.994	0.002
G21	0.995	0.002	0.003	G53	0.002	0.996	0.001
G22	0.989	0.007	0.004	G54	0.003	0.994	0.002
G23	0.992	0.004	0.003	G55	0.003	0.988	0.009
G24	0.990	0.006	0.003	G56	0.002	0.995	0.003
G25	0.979	0.017	0.004	G57	0.002	0.996	0.003
G26	0.992	0.004	0.004	G58	0.002	0.996	0.002
G27	0.794	0.203	0.003	G59	0.003	0.963	0.033
G28	0.988	0.008	0.003	G60	0.003	0.993	0.004
G29	0.958	0.016	0.026	G61	0.010	0.984	0.006
G30	0.993	0.005	0.002	G62	0.009	0.859	0.132
G31	0.945	0.051	0.005	G63	0.107	0.865	0.028
G32	0.681	0.285	0.034				

**Table 5 genes-14-01182-t005:** Expected heterozygosity (He) and *Fst* values of 3 sub-populations.

Subpopulation (K)	Expected Heterozygosity (He)	*Fst*
1	0.452	0.253
2	0.463	0.330
3	0.444	0.244
Mean	0.453	0.276

**Table 6 genes-14-01182-t006:** Genetic differentiation coefficients for three estimated sub-populations.

Sub-Populations (K)	Sub-Population A	Sub-Population B	Sub-Population C
Sub-population A	-	0.492	0.323
Sub-population B	0.492	-	0.566
Sub-population C	0.323	0.566	-

## Data Availability

Data is contained within the article.

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
