# Peer review of "Genetic Diversity and Population Structure in Türkiye Bread Wheat Genotypes Revealed by Simple Sequence Repeats (SSR) Markers"

_genes, 2023, doi:10.3390/genes14061182_

Round 1
Reviewer 1 Report
1- From the start point I would like to mention that the papers has no novelty because the methods you used (SSR) in this study is too old
2- Write the abbreviation of each Molecular methods
Random amplified polymorphic DNA (RAPD), amplified fragment length polymorphisms (AFLP), inter simple-sequence repeat (ISSR), start codon targeted markers (SCoT ), iPBS-retrotransposons , expressed sequence tag, single nucleotide polymorphism (SNP), next generation sequence (NGS)
3- In the M&M section why you used only 5 genotypes from 63 genotypes to select polymorphic primers?
4- Dose the 651 SSR markers covering all the wheat genome (A, B and D) 21 wheat chromosome and the selected 120 SSR also covered the wheat genome or not?
If yes please mention how many markers in each genome?
5- Could you please write the GD and PIC formula in the M&M section
6- You must add some information about the method which you used in Structure software like how many K values, number of iteration, burn degree and the Markov chain Monte Carlo (MCMC) replications.
7- In the Discussion part (Present PIC values varied between 0.031 - 0.0.915 with an average value of 0.430. In this study) I think there is error it should be (PIC) ranged from 0.031 to 0.915 with a mean 0.43
8- You must explain the biological mean of the 3 sub population meaning that why a set of different genotypes related to each other in the same sub population
Extensive editing of English language required
Author Response
Responses to Comments of Reviewer 1
The author thanks the potential reviewer for her/his valuable time and for considering the manuscript. In the revised MS, we have corrected the text based on all comments and suggestions, as well as all changes have been highlighted in yellow.
We hope the revised version of MS will be merit to accept for publication in the Genes journal.
Sincerely,
Dr. Kamil HaliloÄŸlu
Comments
Comment 1# From the start point I would like to mention that the papers have no novelty because the methods you used (SSR) in this study is too old.
Response: We thank the potential reviewer to consider our work. As it has been denoted, SSR markers are one of the most important molecular markers due to their potential in showing the level of polymorphism in the conserved regions in the plant genome. On the other hand, in numerous studies, this marker technique is commonly used to complete information obtained by new techniques such as SNP and even DarT. It is worth noting that the main objective and also novelty of the present work refers to dissecting the genetic diversity and population structure in a set of Turkish bread wheat genotypes. Indeed, the evaluated genotypes consist of an important part of wheat germplasm in Turkey and our results can help breeders to enrich the genetic background of this important cereal through new breeding programs toward the improvement or development of new varieties.
Comment 2# Write the abbreviation of each Molecular methods; Random amplified polymorphic DNA (RAPD), amplified fragment length polymorphisms (AFLP), inter simple-sequence repeat (ISSR), start codon targeted markers (SCoT), iPBS-retrotransposons, expressed sequence tag, single nucleotide polymorphism (SNP), next generation sequence (NGS).
Response: We have showed the full sentences for each abbreviation.
Comment 3# In the M&M section why, you used only 5 genotypes from 63 genotypes to select polymorphic primers?
Response: As you know, one of the first steps in each molecular study is screening the polymorphic markers. In this task, researchers usually test a large number of markers on a few genotypes or accessions. In this way, the polymorphic markers will be identified, and afterward, all genetic materials screening using them. In the present study, we tested a total of 425 SSR primers through five wheat accessions. Of these, the 120 primers showed the highest rate of polymorphism.
Comment 4# Dose the 651 SSR markers covering all the wheat genome (A, B and D) 21 wheat chromosome and the selected 120 SSR also covered the wheat genome or not? If yes please mention how many markers in each genome?
Response: The additional inforamtion have been added to a supplementary file.
Comment 5# Could you please write the GD and PIC formula in the M&M section?
Response: We thank the reviewer to highlighting this topic. It is worth noting that these parameters have been estimated using the PowerMarker software.
Comment 6# You must add some information about the method which you used in Structure software like how many K values, number of iterations, burn degree and the Markov chain Monte Carlo (MCMC) replications.
Response: We have presented an additional details as you suggested.
Comment 7# In the Discussion part (Present PIC values varied between 0.031 - 0.0.915 with an average value of 0.430. In this study) I think there is error it should be (PIC) ranged from 0.031 to 0.915 with a mean 0.43.
Response: We thank the reviewer to highlighting this mistake. We have corrected it in the revised text.
Comment 8# You must explain the biological mean of the 3-sub population meaning that why a set of different genotypes related to each other in the same sub population.
Response: As we mentioned in several parts of the manuscript, the tested materials have a different genetic background and this issue cases creating a diverse population. Indeed, the grouping of the genotypes in different subpopulations refers to their different genetic background.
Comment 9# Extensive editing of English language required.
Response: The manuscript have revised by a native person.

Reviewer 2 Report
The manuscript of TürkoÄŸlu et al. investigates molecular diversity and population structure of a set of 63 Turkish bread wheat using 120 SSR markers. I am not sure what is the novelty and/or originality behind this study since a rather small population is analyzed with not that many markers (compared to the studies conducted nowadays). Despite not seeing the novelty of this study, I decided to give authors the chance to convince me their findings are of some importance to the field. However, that did not happen. First of all, I noticed that authors did not follow the MDPI Genes instructions for authors – sections are not numbered properly, tables are not described and refereed to properly, they did not pay attention to use a proper citation style when describing references, etc. Moreover, the Latin name of the species investigated is not correct in more than a half of the MS. I am aware these flaws are easy to correct but they all together give an impression of poorly written MS and like authors did not put a lot of effort into it. For me as a reviewer it was genuinely difficult to follow some results and explanations since tables are not cited properly. Is it my job to read authors’ minds and to know which table are they actually referring to? Additionally, in some sections (especially Introduction) English is very difficult to understand. It seems like some sentences got lost in translation or are not finished at all (I left some comments in the pdf attached). Thus, extensive editing of English would be required. Although being full of these “technical” issues, which can be corrected, this MS is also lacking a true purpose.
Introduction does provide sufficient background, which was not that difficult to accomplish due to the rather narrow area this MS covers. However, authors should be more specific when setting research aims. The aim of this study remained completely unclear to me and I missed the point/purpose of this study. How and why are present findings expected to form a basis for further breeding programs? Why is determining population structure of chosen varieties important for this goal? Material and Methods section is in general well-written but I remained confused by such a high number of different software used in this study. Results are in general clearly presented, but sometimes difficult to follow since tables are not cited properly. Almost the whole Discussion section is written mainly as a literature overview without connecting it with the results of the present study. In fact, the results of the present study are barely discussed at all. Why are these findings important? How are they going to be used in future breeding process? Since none of these was actually discussed in the Discussion section I expected the Conclusion section will explain in more details the importance of these findings. However, conclusion is in large part just the repeated introduction (DNA markers etc.) together with basic results (the highest GD observed etc.). So the only real conclusion of this study is that the number of cultivars or polymorphic markers should be increased?
For specific comments please check the pdf attached.

In some sections (especially Introduction) English is very difficult to understand. It seems like some sentences got lost in translation or are not finished at all (I left some comments in the pdf attached). Thus, extensive editing of English would be required.
Author Response
Responses to Comments of Reviewer 2
We thank the potential reviewer for her/his time and forward valuable comments on our manuscript. We have tried addressed to all comments and suggestion in the revised text.
Sincerely,
Dr. Kamil HaliloÄŸlu
Comments
General comment #
The manuscript of TürkoÄŸlu et al. investigates molecular diversity and population structure of a set of 63 Turkish bread wheat using 120 SSR markers. I am not sure what is the novelty and/or originality behind this study since a rather small population is analyzed with not that many markers (compared to the studies conducted nowadays). Despite not seeing the novelty of this study, I decided to give authors the chance to convince me their findings are of some importance to the field. However, that did not happen. First of all, I noticed that authors did not follow the MDPI Genes instructions for authors – sections are not numbered properly, tables are not described and refereed to properly, they did not pay attention to use a proper citation style when describing references, etc. Moreover, the Latin name of the species investigated is not correct in more than a half of the MS. I am aware these flaws are easy to correct but they all together give an impression of poorly written MS and like authors did not put a lot of effort into it. For me as a reviewer it was genuinely difficult to follow some results and explanations since tables are not cited properly. Is it my job to read authors’ minds and to know which table are they actually referring to? Additionally, in some sections (especially Introduction) English is very difficult to understand. It seems like some sentences got lost in translation or are not finished at all (I left some comments in the pdf attached). Thus, extensive editing of English would be required. Although being full of these “technical” issues, which can be corrected, this MS is also lacking a true purpose. The manuscript of TürkoÄŸlu et al. investigates molecular diversity and population structure of a set of 63 Turkish bread wheat using 120 SSR markers. I am not sure what is the novelty and/or originality behind this study since a rather small population is analyzed with not that many markers (compared to the studies conducted nowadays). Despite not seeing the novelty of this study, I decided to give authors the chance to convince me their findings are of some importance to the field. However, that did not happen. First of all, I noticed that authors did not follow the MDPI Genes instructions for authors – sections are not numbered properly, tables are not described and refereed to properly, they did not pay attention to use a proper citation style when describing references, etc. Moreover, the Latin name of the species investigated is not correct in more than a half of the MS. I am aware these flaws are easy to correct but they all together give an impression of poorly written MS and like authors did not put a lot of effort into it. For me as a reviewer it was genuinely difficult to follow some results and explanations since tables are not cited properly. Is it my job to read authors’ minds and to know which table are they actually referring to? Additionally, in some sections (especially Introduction) English is very difficult to understand. It seems like some sentences got lost in translation or are not finished at all (I left some comments in the pdf attached). Thus, extensive editing of English would be required. Although being full of these “technical” issues, which can be corrected, this MS is also lacking a true purpose.
Response: We thank the potential reviewer for her/his time and helps us to improve manuscript. As we mentioned in the last part of the Introduction section, the main objective of the present study was to investigate the genetic diversity in a set of released genotypes in Turkish bread wheat. Since the genetic basis of improved genotypes is narrowed over a long time due to repeated use and breeding cycles, we surmise that the results of the present work can be useful for Turkish wheat breeders to exploit suitable genotypes for use of them in the hybridization programs. In general, in the revised text, we have corrected all grammatical errors, re-write missing sentences, re-check references list according to journal’s style, and corrected the captions for all tables and figures. Moreover, the highlighted parts in the Introduction and discussion sections have improved. Furthermore, the conclusion section has corrected. To polish the language of the manuscript, we have requested a native person to do it, and all corrections have highlighted by track-changes. In general, all changes based on your comments have highlighted in red. I hope the revision version of manuscript is acceptable for publication.

Reviewer 3 Report
There is urgent requirement to improve wheat genetic resources to meet global challenges of meeting increased food demand and worsening global climate. In present study, authors have performed molecular diversity and population structure analysis of Turkish bread wheat using 120 SSR markers. The data revealed a number of polymorphic alleles useful for genetic characterization; suggesting the studied genotypes possess an organized population structure. Authors have performed several significant statistical analyses.
Overall, the manuscript presents a timely report on genotyping of Turkish bread wheat. The results provide supportive data for further studies on wheat genetic resource management for plant breeding programs. The manuscript could be considered for publication, after careful revision of current draft.
Some comments:
· In table 1 please specify what is mean by alternative sowing?
· For each Figure caption, please elaborate the captions and describe the content of each figure in brief so that it is easy to follow to readers.
· The abbreviations should be explained before they are used.
· Author need to enrich the scientific literature of molecular markers and its usage in plant breeding.
· See below few examples from literature on molecular marker-based analysis (using RAPD, RFLP, AFLP, SSR, ISSR, ITS, etc):
· Multiplex molecular marker-assisted analysis of significant pathogens of cotton (Gossypium sp.), 2022; Biocatalysis and Agricultural Biotechnology https://doi.org/10.1016/j.bcab.2022.102557 (Cotton); Microsatellite and RAPD analysis of grape (Vitis spp.) accessions and identification of duplicates/misnomers in germplasm collection, Upadhyay et al., 2010 Indian J Hortic Volume 67 Pages 8-15;
· In conclusions, please include key statistical results to emphasize the significant findings of current study.
Content language is good.
Author Response
Responses to Comments of Reviewer 3
The authors would like to thank the potential reviewer for her/his valuable comments. In the revised text, we have addressed to all comments point-by-point and highlighted them in green.
Sincerely,
Dr. Kamil HaliloÄŸlu
Comments
Comment 1# There is urgent requirement to improve wheat genetic resources to meet global challenges of meeting increased food demand and worsening global climate. In present study, authors have performed molecular diversity and population structure analysis of Turkish bread wheat using 120 SSR markers. The data revealed a number of polymorphic alleles useful for genetic characterization; suggesting the studied genotypes possess an organized population structure. Authors have performed several significant statistical analyses. Overall, the manuscript presents a timely report on genotyping of Turkish bread wheat. The results provide supportive data for further studies on wheat genetic resource management for plant breeding programs. The manuscript could be considered for publication, after careful revision of current draft.
Response: We thank the reviewer for her/his positive feedback on our work.
Comment 2# In table 1 please specify what is mean by alternative sowing?
Response: We have changed it with a suitable word.
Comment 3# For each Figure caption, please elaborate the captions and describe the content of each figure in brief so that it is easy to follow to readers.
Response: We thank the reviewer to highlighting this issue. We have presented a complete information in each caption.
Comment 4# Author needs to enrich the scientific literature of molecular markers and its usage in plant breeding.
Response: We have updated the literatures.
Comment 5# In conclusions, please include key statistical results to emphasize the significant findings of current study.
Response: We have improved this section.
Comment 6# Content language is good.
Response: We thank to the potential reviewer to her/his positive feedback.

Round 2
Reviewer 1 Report
the manuscript was improved
I suggest its publication
English was improved